# Bioactive Agent-Loaded Electrospun Nanofiber Membranes for Accelerating Healing Process: A Review

**DOI:** 10.3390/membranes11090702

**Published:** 2021-09-13

**Authors:** Seyyed-Mojtaba Mousavi, Zohre Mousavi Nejad, Seyyed Alireza Hashemi, Marjan Salari, Ahmad Gholami, Seeram Ramakrishna, Wei-Hung Chiang, Chin Wei Lai

**Affiliations:** 1Department of Chemical Engineering, National Taiwan University of Science and Technology, Taipei 106, Taiwan; kempo.smm@gmail.com; 2Biotechnology Research Center and Department of Pharmaceutical Biotechnology, Faculty of Pharmacy, Shiraz University of Medical Science, Shiraz 71345-1583, Iran; zohreh.tnt@gmail.com (Z.M.N.); gholami@sums.ac.ir (A.G.); 3Nanomaterials and Polymer Nanocomposites Laboratory, School of Engineering, University of British Columbia, Kelowna, BC V1V 1V7, Canada; s.a.hashemi0@gmail.com; 4Department of Civil Engineering, Sirjan University of Technology, Sirjan CM7X+MCX, Iran; salari.marjan@gmail.com; 5Center for Nanofibers and Nanotechnology, Department of Mechanical Engineering, National University of Singapore, Singapore 117576, Singapore; seeram@nus.edu.sg; 6Nanotechnology and Catalysis Research Centre (NANOCAT), Institute for Advanced Studies (IAS), University of Malaya (UM), 50603 Kuala Lumpur, Malaysia

**Keywords:** wound dressing, bioactive molecules, wound healing, electrospinning, tissue engineering

## Abstract

Despite the advances that have been achieved in developing wound dressings to date, wound healing still remains a challenge in the healthcare system. None of the wound dressings currently used clinically can mimic all the properties of normal and healthy skin. Electrospinning has gained remarkable attention in wound healing applications because of its excellent ability to form nanostructures similar to natural extracellular matrix (ECM). Electrospun dressing accelerates the wound healing process by transferring drugs or active agents to the wound site sooner. This review provides a concise overview of the recent developments in bioactive electrospun dressings, which are effective in treating acute and chronic wounds and can successfully heal the wound. We also discuss bioactive agents used to incorporate electrospun wound dressings to improve their therapeutic potential in wound healing. In addition, here we present commercial dressings loaded with bioactive agents with a comparison between their features and capabilities. Furthermore, we discuss challenges and promises and offer suggestions for future research on bioactive agent-loaded nanofiber membranes to guide future researchers in designing more effective dressing for wound healing and skin regeneration.

## 1. Introduction

Skin plays an essential role in protecting our body against any physical and chemical injury and water loss, and contributes to the maintenance of bodily homeostasis [1,2,3]. After damage, the skin should restore its integrity to maintain its functions. Wound healing is a vital and complex physiological process consisting of the collaboration of many cell strains and their products [4]. In this process, the growth factors induce cell proliferation, which leads to a combination of several dynamic changes, including soluble mediators, blood cells, the production of the extracellular matrix, and the proliferation of parenchymal cells [5,6]. In healthy skin, the outer layer is the epidermis, which is impermeable and resists the external environment. The dermis, the inner layer of skin, is full of extracellular matrix (ECM), mechanoreceptors, and vasculature, and this layer can also provide the skin with immunity, nutrients, and mechanical strength [6,7].

In general, a wound is a type of injury that involves damage to the epithelial tissue and causes a lack of skin integrity. Wound healing is a natural physiological response to tissue damage. In general, wound healing is performed with four specific stages of hemostasis, inflammation, proliferation, and maturation (Figure 1) [8,9,10]. ECM plays a crucial role in every step of the wound healing process. The ECM acts as a supporting scaffold, which is necessary at all stages of the repair phenomenon [11]. The role of ECM is not only in the activity of cytokines and signaling within cellulitis, but also in mechanical support, which is helpful for activating and regulating the pathways of cell differentiation and proliferation [12,13].

Successful tissue regeneration depends on the complex function and fibrous form of native ECM stimulated by skin substitutes or wound dressing membranes [14]. Various biocompatible and biodegradable scaffolds have been created with three-dimensional cell proliferation as a suitable dressing membrane for wound healing [15,16]. Extensive studies in the field of tissue engineering have been performed to define tissue engineering as described in the workshop of the National Science Foundation, and the role of scaffolds in maintaining and improving tissue function and reconstruction is undeniable [17].

**Figure 1 membranes-11-00702-f001:**
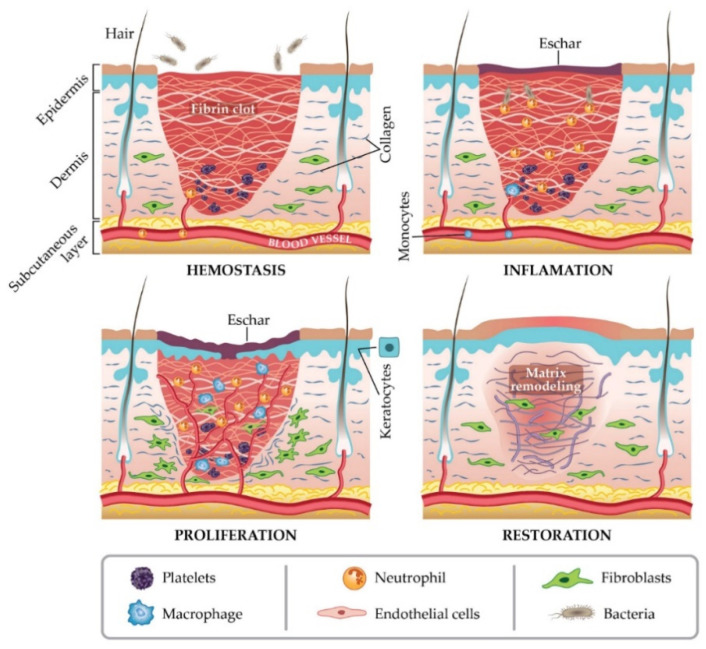
Stages of wound healing. Adapted from I. Negut, V. Grumezescu, A.M. Grumezescu, Treatment strategies for infected wounds, adapted from [18].

Electrospinning is a simple and effective technique that uses electric force to produce nano-/micro-scale continuous fibers made of a polymer solution Figure 2A. The nanofibrous membrane has impressive advantages as skin scaffolds compared to conventional and non-fibrous structures such as hydrogels, due to their similarity to native ECM of skin tissue, so they can be considered as potential scaffolds providing cell adhesion and proliferation, and eventually induce neodermis regeneration [19,20,21].

To date, the electrospinning method is the most practical technique in the field of advanced scaffolds for tissue engineering and delivery of bioactive agents to damaged tissue, which has attracted the attention of researchers [8]. One of the most practical electrospinning methods in tissue engineering is skin regeneration [22,23,24,25,26,27]. Synthetic polymers are not inherently bioactive, and by combining with some active components, the bioactivity of synthetic polymers can be improved [28]. The electrospinning technique allows the production of a suitable nanofibrous membrane to deliver bioactive agents or pharmacological agents to the damaged site of skin to accelerate the skin regeneration process [14,29]. Bioactive incorporation into electrospun nanofiber membrane could be done in four different ways represented in Figure 2B–E.

**Figure 2 membranes-11-00702-f002:**
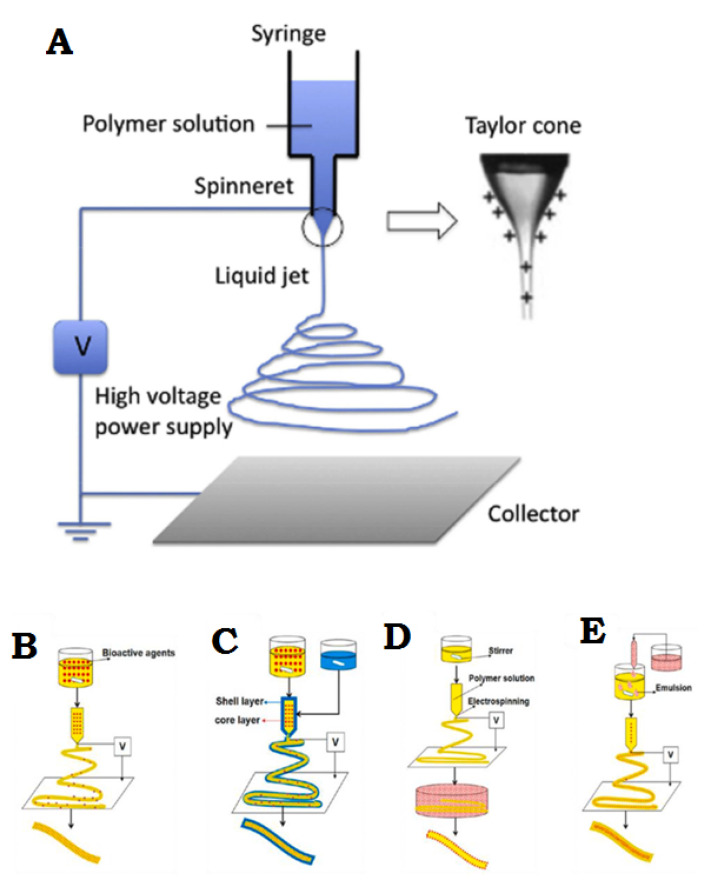
(**A**) General flowchart of the basic electrospinning process. (**B**) Adjusting the electrospinning mixture. (**C**) Set up of shell electrospinning. (**D**) Surface functional electrospinning settings. (**E**) Emulsion-based electrospinning operation. (**A**) is reprinted from [30]; (**B**–**E**) are reprinted from [31] Licensed by the American Chemical Society.

Here in this review article, an attempt is made to collect and present the latest available developments and information on the development of ECM simulator membranes for wound healing applications. Additionally, new biologically active agents used in combination with electrospun wound dressings to improve their therapeutic potential in wound healing are investigated. In addition, here we present commercial dressings loaded with bioactive agents with a comparison between their features and capabilities. Furthermore, we discuss challenges and promises and offer suggestions for future research on bioactive agent-loaded nanofiber membranes to guide future researchers in designing more effective dressing for wound healing and skin regeneration.

## 2. New Advances in Wound Dressing of Bioactive Nanofibers

A new generation of wound dressings with the ability to quickly accelerate the wound healing process is nanofiber membranes with active materials. The production of this type of membrane can be performed using any kind of electrospun solution. In general, a biologically active factor can be considered to play an active role in the recovery process. Bioactive agents include antibacterial particles, growth factors, stem cells, vitamins, and other factors that are effective in wound healing in combination with nanofiber membranes.

In a new study, for the first time, a hybrid nanofiber membrane was proposed as a bioactive wound dressing by Homaygohar et al. The results of this study showed that biohybrid nanofibers have elastic modulus and elongation (300% and 58%, respectively) comparable to regular poly(ε-caprolactone) (PCL) nanofibers. Survival of WST-8 cells, and NIH3T3 fibroblasts proliferating correctly and interacting with biohybrids were observed in the experimental results. It was also concluded from SEM images that cells can adhere to such nanofibers even better than pure PCL cells. As a result, sufficient traction and mineralization through this BSA/PCL nanofiber biohybrid membrane can accelerate the wound healing process and be considered a suitable option for wound dressing applications [28].

In another study, Cerium-doped bioactive glass (Ce-BG)-loaded chitosan/polyethylene oxide (PEO) nanofibrous scaffolds were fabricated as antibacterial wound dressing. Chitosan/PEO scaffolds had been modified to overcome the instability and mechanical properties problems of these scaffolds in aqueous solutions. Numerous parameters including mechanical properties, antibacterial activity against Gram-positive and Gram-negative bacterial cells, ion release behavior, and swelling properties were studied in this research. This study shows that increasing the Ce-BG/CH (*w*/*w*) ratio up to 20% (*w*/*w*) can increase the swelling and mechanical properties of scaffolding. By further increasing the ratio, the effects of Ce-BG affinity on the above are reduced. In addition, chitosan/PEO scaffolds containing Ce-BG had mechanical properties very close to those of the skin, and the results showed that they were only 20% shorter than the required length when breaking skin tissue scaffolds, which increased in length by 28.6% when broken [32].

Another interesting study was designed and conducted using Lemon Oil (LMEO) as a natural and highly antibacterial bioactive agent for making electrospun cellulose acetate (CA) nanofiber membranes by Bekzadeh et al. The chemical, antibacterial, morphological, thermal and optical properties of LMEO-loaded CA nanofiber scaffolds were investigated.

The degradation profile of CA with LMEO composition showed no change in the results of the calorimetric analysis. LMEO-loaded CA electrospun nanofibers 100% eliminated Escherichia coli and Staphylococcus aureus in the antibacterial evaluation, even at the lowest LMEO loading concentration. LMEO electrospinning fiber membranes show that nanofibers can retain their antibacterial properties even after two months of storage. Consequently, this nanofiber structure may be a potential candidate as a bioactive wound dressing with long-lasting antibacterial activity [33].

## 3. Use of Bioactive Agents in Combination with Electrospun Fiber Binders

Nanocomposites are of two main categories: organic and organic mixtures, and organic and inorganic mixtures; these are described in the following sections.

### 3.1. Organic–Organic Mixtures

Organic and organic compounds of synthetic and natural polymers are created with the main aim of improving biological activities and functions. Table 1 summarizes the organic and organic mixtures of nano-organic scaffolds discovered by various researchers. The first groups to use the composite concept to develop nanobiometric and bioactive scaffolds proposed the efficiency of using a biocompatible collagen-derived compound. Some specimens that use gelatin to obtain physical, chemical, and biological properties include polymeric gelatin (Gt) combined with synthetic PCL [34]. Our results showed that ferrous nanomaterials equal to Gt/PCL have very balanced mechanical properties and moisture in water compared to compounds. In vitro cell culture experiments showed significant cell proliferation and degradation compared to synthetic PCL scaffolds. Cells were first measured by laser scanning microscope, showing the Gt/PCL composite nanotube scaffolds to be up to 110 μm in size. The introduction of Gt-activated biopolymer to PCL results in significant wetting and cell viability in bone scaffolds and relative cellular responses to the combined effect of the material.

The nanoelectrospray electrical scaffolds created by the shortcuts are porous, and the “pores” formed are much smaller than the size of normal cells, at up to ten microns, which can prevent cell migration to the nanoelectrospun structure. There are three causes of cell coagulation.

First, natural biopolymers introduce hydrophilic/moisturizing signals and good biological, cognitive signals from Gt to PCL, thus facilitating the transfer of nutrients/oxygen and the elimination of metabolism. A favorable local microenvironment such as the constituent results of the material can undoubtedly modulate the cellular reaction behavior. This encourages pioneering products and cells to migrate in the scaffold [35,36]. Second, Gt/PCL composites have excellent tensile and deformation properties, but lower tensile strength. One of the essential factors of cell penetration is the nanomechanical properties of the composites. These favorable mechanical properties can open spaces allowing cell penetration to deep scaffold surfaces. The flexibility and deformation of nanoscaffolds, on a large scale, affect the laboratory results and morphology of cells [37].

Over time, the gelatin in the Gt/PCL scaffold becomes porous and permeable, and gradually dissolves during cell culture. This facilitates the easy transfer of nutrients and waste products and creates more space for cell migration. The formation of three-dimensional porous materials by washing the gelatin components from the composite materials was shown in a subsequent study [38]. The three-dimensional porosity morphology shows that the gelatin phase separation and PCL in the nanocomposite compounds are randomly mixed as presented in Figure 3. BET surface measurements showed that 3-D porosity had an area of about 2.4 times the size of the Gt/PCL without porosity. On the basis of these results, an electrospun Gt/PCL composite nanoparticle was recently developed on a polyurethane coating (Tegaderm ™, 3M Medical, Two Harbors, MN, USA) for potential application in skin wound healing.

Through regeneration and adhesion, the growth and proliferation of cells in the structure of Tegaderm-nanofibers indicates high potential and feasibility in wound healing.

**Table 1 membranes-11-00702-t001:** Organic–organic composite material with nanoscaffolding.

Scaffold Materials	Solvents Used	Tissue Engineering Applications	Cells Cultured	Diametes of Electrospun Fibers	References
Gelatin/PCL	TFE	Skin	BMSCs, Fibroblasts	500–900 nm	[34]
Collagen/Elastin/PEO	Aqueous HCL	Blood vessel	SMCs	220–600 nm	[39]
Collagen/P(LLA-CL)	HFIP	Blood vessel	HCAECs	100–300 nm	[40]
PDO/Elastin	HFIP	Vascular graft	Human dermal	400–800 nm	[41]
PHBV/Collagen	HFIP	/	NIH3T3	300–600 nm	[42]
Gelatin/Elastin/PLGA	HFIP	Heart/blood	H9c2 rat cardiac	380 + 80 nm	[43]
Gelatin/PANi	HFIP	Cardiac/nerve	Smooth muscle cells H9c2rat cardiac mvoblast	60–800 nm	[44]
DNA/PLGA or PLA-PEG block copolymer	DMF/Tris-EDTA	Bone	Aper osteoblastic cell line MC3T3EI	250–875 nm	[45]
NGF-BSA/PCLEEP	DCM/PBS	Nerve	PCI2 cells	0.5–3 μm	[46]
PGA/Chitin	HFIP	/	Fibroblasts	50–350 nm	[47]

### 3.2. Organic–Inorganic Blends

Table 2 summarizes the organic and inorganic compositions of the nanoparticles, often included in the polymer field to add properties or improve mechanical properties for bone tissue engineering [48,49,50], nanoparticles [51], and mustache [52]. Nanoscaffolding has been reported to prepare bone tissue engineering.

Bone is a natural compound composed of apatite mineral nanocrystals that are an organic matrix (mainly type I collagen). Natural options for bone tissue engineering applications include bone structure reconstruction, hydroxyapatite and other calcium phosphates along with biodegradable and biocompatible polymers.

Fuji Hara et al. [53] reported PCC/CaCO_3_ nanocomposites of polycaprolactone with two types of PCL to calcium carbonate with a ratio of PCCO: CaCO_3_ of 75: 25% by weight and 25: 75% by weight.

A good cell transplant range was observed for the study compound, indicating the potential of using PCL/CaCO_3_ nanocomposites for bone regenerating membrane (GBR).

Similar results have been reported for hydroxyapatite nanoparticles in other polymer systems such as synthetic poly (lactic acid) (PLA) [48] natural polymers (e.g., gelatin) and Silk [48,49,50]. A combination of cellular signal molecules such as RGD peptides and further growth factors has been shown to enhance the cellular behavior of tissue engineering scaffolds. Human embryonic osteoblast (hFOB) caused a significant increase in minerals (55%) in PCB/nHA/collagen biocomposite scaffolds after ten days of cell culture.

Such unique combinations of nanostructures and biological activity in nanoparticle scaffolds effectively act for the adhesion, proliferation and mineralization of hFOB and form bone tissue. Electrical scaffolds containing bone morphogenetic protein 2 (BMP-2), silk nanoparticles, or hydroxyapatite nanoparticles were proposed for in vitro bone formation from human bone marrow mesenchymal stem cells (hMSC) by Lee et al. [38]. The researchers’ results showed that BMP-2 and nHA in electrospun silk plastic nanoparticles lead to more calcium deposition and rearrangement of BMP-2 than other systems.

**Table 2 membranes-11-00702-t002:** Organic-inorganic nanocomposite scaffolds.

Scaffold Materials	Diameters of Electrospun Fibers	Cells Cultured	Solvents Used	Solvents Used	Tissue Engineering Applications	References
HA/Gelatin	200–400 nm	Human osteoblastic cells MG63	HFIP	HFIP	bone	[49]
Silk/PEO/Nhap/BMP-2	520 + 55 nm	hMSCs	Water	Water	bone	[44]
PLLA/HA	500 nm	Human osteosarcoma	DCM/1,4 dioxane	DCM/1,4 dioxane	bone	[54]
PLLA/MWCNT/HA	250–950 nm	DPSCs	DCM	DCM	dental	[54]
PHBV/HAp	100–2000 nm	COS-7 cells from the monkey kidney	TFE	TFE	/	[55]
PCL/CaCO3	760 + 190 nm	Human osteoblastic hFOBI.19	HFIP	Chloroform/Methanol	bone	[53]
HA/PLA	1–2 μm	MG63 cells	HFIP	Chloroform	bone	[48]

It has been reported that nanocomposite compounds and nanoparticles and microstructures and morphology have essential effects on biological responses that ultimately depend on artificial processing.

It has been reported that not only nanocomposite compounds but also nanoparticles and microstructures and morphology have important effects on biological responses that ultimately depend on artificial processing.

To date, simple agitators and ultrasonic cations have been used to disperse nanocomposites further to make nanoparticles with polymers. Large area accumulation of nanoparticles and surface interactions of nanoparticles is one of the problems of their accumulation and processing.

NHA particle sizes range from 10 nm to 150 nm. Non-spherical nanoparticles such as HA or carbon nanotube (CNT) nanoparticles do not have the required direction and balance in nanocomposites and also do not have a uniform distribution of nHA in the polymer matrix.

Unlike native ECM, reconstruction of such nanocomposites is time consuming. The surface forces between nanoparticles and polymers must be carefully considered to solve this problem. In another study, Kim et al. (2006) [48] suggested the use of hydroxysteric surfactant (HSA) to control the interaction between hydrophilic powders nHA and hydrophobic PLA dissolved in hydrophobic chloroform. By improving the dispersion of nHA powders, they led to the uniformity of nanocomposite materials. The biometric method for mimicking the structures and compositions of human tissues should be considered one. The diameter (1–2 μm) is still relatively large, a common feature for electrical components produced from leached nanoparticles [56].

### 3.3. Commercialized Bioactive Wound Dressing

Today, there has been a significant increase in the production of wound care medical products, so that new dressing methods with many capabilities for wound care are being developed. According to patent offices, Arvia has introduced thousands of patents for wound coatings filled with bioactive substances. Previous research has been conducted to introduce two patents invented in Poland (PL 236368, PL 236367) describing the production methods of Cordlan/foam agarose dressings and hydrogels (Figure 4). Both types of ulcers can be produced by adding active ingredients such as vitamins and growth factors.

The means for making an active wound dressing using chitosan and curcumin using electrostatic rotation is described by Chinese inventors in the specification CN107475812A.In this invention, the porous structure of a biological substance with antibacterial (*E. coli*) and anti-inflammatory properties was introduced. Due to the very specific level of biological material, it can be loaded with multiple drugs, acting as a Chinese drug delivery system. Another Chinese patent, numbered CN110025817A, is a method of preparing a fiber dressing. This provides antibacterial composites containing essential oils. The thiol dressings for bioactive wounds obtained are characterized by good permeability and antibacterial properties against *E. coli* and *S. aureus*.

The invention described in German patent number 10426670 relates to wound dressing. This wound dressing has a porous structure and/or foam and an adsorbent in the form of activated carbon with at least one layer permeable to air. Structurally, it is largely in the form of a solid foam (“foam layer”). https://patents.justia.com/patent/10426670 (accessed on: 1 July 2021).

Taiwanese patent number TW201208717A introduces the method of producing a wound-activating dressing containing citrus extract as a new product. The method of producing citrus-based wound dressings is based on the wet rotation method or the soaking method. The increase in cell proliferation and angiogenesis based on the production of an antibacterial film is described in another Chinese patent under CN105963753A. This explains the method of producing bioactive wounds based on chitosan and curcumin using electrostatic rotation and biomaterials described in the invention of multilayer dressings containing natural plant ingredients (marigold oil, Chinese registered tea with specification number CN107475812A).

This method introduces the porous structure of a biometric material with antibacterial (*E. coli*) and anti-flexibility properties. The special feature of biometric materials is the very special surface of these materials as a drug delivery system.

Another Chinese patent introduces a method for preparing an antibacterial compound containing essential oil with the specification number CN110025817A.

Wound dressings obtained using bioactive materials have significant properties including good permeability and antibacterial properties against *E. coli* and *S. aureus*. A Taiwanese patent describes a method for producing a wound-activated dressing containing citrus extract with the specification number TW201208717A.

The method for producing wound dressings is based on the method of wet rotation or the method of soaking with the citrus extract content. Additionally, experiments on the biological material of chitosan rich in citrus extracts demonstrated increased cell proliferation and angiogenesis. Another Chinese invention is the production of an antibacterial film, with the specification number CN105963753A.

The patented biological material is a multi-ingredient dressing containing natural plant materials (marigold oil, tea polyphenols, red oak powder and marigold oil), which is described as non-toxic, with good elasticity and disinfectant properties.

The biomaterial results presented in this paper are not currently used in practice. However, many active wound dressings are used in global markets as medical wound care. Several companies offer wound care products, producing dressings enriched with a substance that accelerates the reproductive process.

The German company Hartmann HydroClean introduced a super-absorbent wound dressing. This adsorbent contains polyhexane (PHMB), and the dressing is activated with Ringer’s solution. Due to the mentioned compounds, the dressing has antimicrobial properties and accelerates the healing process with the help of ginger-like fluids released in the wound bed [58].

A British company, Advancis Medical, has introduced an algae alginate sleeve with Manuka honey. The antibacterial properties of this compound are provided by glucose oxide in the body. The biomaterials presented in this paper are in the research phase and are not currently used in practice. However, many active wound dressings are now available in global markets that are mostly used as medical wound care products. Among companies providing wound care materials, some are producing dressings enriched with a substance that accelerates the reproduction process.

The German company Hartmann HydroClean introduced a dressing that, in addition to the super-absorbent wound dressing, is activated with Ringer’s solution and contains polyhexane (PHMB). This dressing has antimicrobial properties and accelerates the healing process with the help of rust material that is released in the wound bed [58].

Antibacterial properties are guaranteed by the glucose oxidase in honey. This enzyme facilitates the formation of hydrogen peroxide in the wound bed with an antiseptic and prevents biological formation. In addition, the biological substance ensures a moist environment and cleanses the arteries. Algion is assigned to various types of wounds [59]. Recently, commercial dressings containing silver particles have been used to control a wide range of microorganisms in wound ulcers. Mepilex Border Ag floor dressing is an example of a commercial antimicrobial dressing containing molasses silver particles. This silicone care dressing contains activated carbon, silver sulfate and a Safetac wound contact layer. This substance is assigned to moderate and highly inflamed wounds such as partial thickness burns, surgical and traumatic wounds, pressure or leg wounds [60]. Biatain Ag dressing is distributed in a silver structure. This new product also provides care for severe wounds. As a result of contact between this product and the wound, silver is released after 7 days. Extensive studies conducted by researchers on this dressing have concluded that the dressing is very effective for combatting bacterial infections [61]. Another manufacturer, Coloplast, introduced a dressing of carboxymethylcellulose, calcium alginate and silver ions. One of the characteristics of this dressing, called iatain^®^ Alginate Ag, in addition to its antibacterial effect, is its hemostatic effect on the wound site [62]. Other commercial dressings include ACTICOAT ™, PolyMem WIC Silver^®^, Suprasorb^®^ A + Ag, and Atrauman^®^ based on silver nanoparticles commonly used to treat infectious wounds [63].

## 4. Challenges and Future Perspectives

Novel strategies for combining electrospun nanofibers with bioactive agents need to be created to obtain the best bioactive electrospun wound dressing in order to accelerate the wound treatment process. Different bioactive features can be obtained, for example, by blending, powder impregnation, electrospray of natural and synthetic drugs, taking advantage of biowaste and nutraceuticals. For the development of a specific application, the fundamentals of electrospinning process need to be comprehended. For instance, it is essential to understand which one of electrospinning techniques must be used to fabricate nanofibers in conjugation with a determined bioactive agent. Different bioactive agents and biocompatible electrospinnable materials must be studied to achieve the best bioactive electrospun structure. In the future, it is expected that the clinical efficacy of the recent developments presented in Section 2 will be tested and their effect on the wound healing process evaluated. However, the challenge of mass production of these bioactive nanofibrous wound dressings is still a barrier for transferring them from the lab to the clinic.

## 5. Conclusions

In recent decades, significant progress has been demonstrated in the development of novel and effective solutions for skin tissue engineering. Currently, there is a great interest in electrospun membranes due to their specific mechanical and biological properties. Nanofibrous membranes are a potential candidate for delivering bioactive agents to the wound site and among the skin substitutes, bioactive wound dressings are one of the most efficient and most attractive groups.

In this review, we provide a concise overview of recent developments in bioactive electrospun wound dressings that are able to accelerate both acute and chronic wound treatment process. Moreover, we gave an account of bioactive agents with a focus on organic and inorganic bioactive materials used in combination with electrospun wound dressings to improve their therapeutic potential in wound healing. In addition, patents and commercial dressings loaded with bioactive agents were also covered in this review paper.

## Figures and Tables

**Figure 3 membranes-11-00702-f003:**
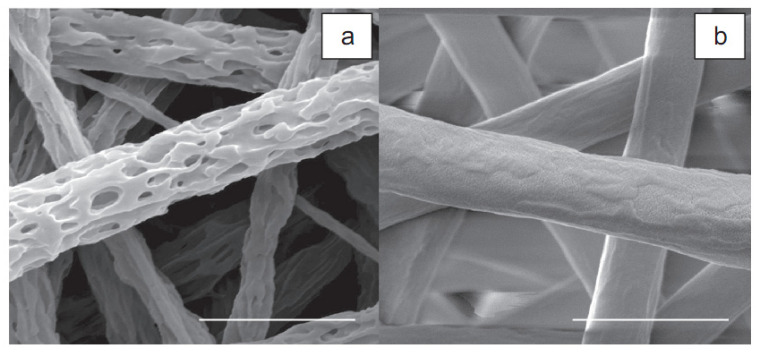
SEM images of 3-D porous material: (**a**) after gelatin is removed from the composite material electrospun Gt/PCL; (**b**) 2 μm scale tape adapted with permission from [38].

**Figure 4 membranes-11-00702-f004:**
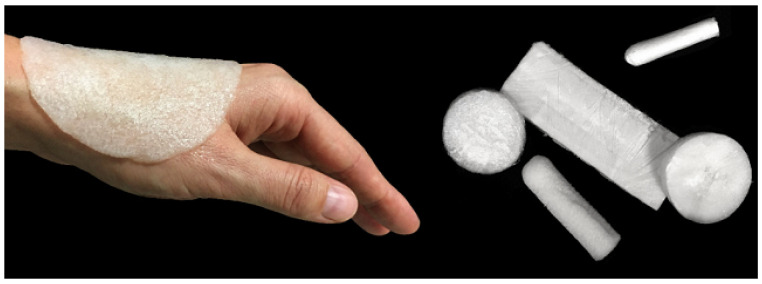
Agarose/curd dressing materials such as foam in different shapes and sizes according to the Polish patent number PL 236,367 [57].

## Data Availability

Not Applicable.

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
