# Peer review of "Bioactive Agent-Loaded Electrospun Nanofiber Membranes for Accelerating Healing Process: A Review"

_membranes, 2021, doi:10.3390/membranes11090702_

Round 1
Reviewer 1 Report
The paper is welldone giving sufficient on the topic treated.In my opinion it is necessary to report each tpbibliographic reference by at least 5 authors before et al. ,according to the international rules.
Author Response
|
Thank you very much for reviewer’s suggestion. The reference’s format has been updated and revised accordingly. |

Reviewer 2 Report
Mousavi and coauthors reported on a review paper addressing bioactive nanofiber loaded membranes to accelerate healing. The work is interesting and timely. Beside correcting a number of typos, a few main points needs to be revised. The review is short and could be considered a mini review.
Major comments
Q1: Introduction: “Synthetic polymers have poor biocompatibility” this is not completely true in skin damages and for sure it is false in a broad sense, you may think about biodegradable polyesters (PLLA, PLGA, PCL) for example, which offer remarkable examples of very good outcomes after implantation in many body settings. Biocompatibility is in fact the ability to perform without adverse host reaction in a specific tissue/organ condition. I suggest either to change with “Synthetic polymers are not inherently bioactive”.
Q2: “In this research work, an attempt is made to collect and present the latest available developments and information on the development of ECM simulator membranes for wound healing applications.” This is not a research work, is a review of existing literature! Please reformulate in a proper context.
Q3: Section 3: patents should be listed in a table including not only Chinese patent numbers. E.g., refer to patent numbers for all, including also for German and British etc.
Q4: Section 3.4. Challenges and Future Perspectives is not effective. Authors must improve this section by mentioning that different bioactive features can be obtained for example by blending, powder impregnation, electrospray of natural and synthetic drugs, taking advantage of biowaste and nutraceuticals. To clarify, I write hereafter some examples that the authors can evaluate, but others are welcome. I intend that the authors must give explicitly proposals of what they think would be the future trends, also taking ideas from neighboring emerging fields and identify new interesting routes. For example, the authors can mention about adding sensing ability to the wound dressings to monitor pH or other parameter as a healing sensor (e.g., intelligent or smart wound dressings). This would be a method to evaluate if healing is accelerated.
Minor comments
Introduction:
“cells [5,6].In healthy skin,” a blank space is needed after punctuation.
“is full of ECM”, the extended name is needed the first time an acronym is mentioned in the text. Abstracts stands as a separate item.
“and strength [6,7]”, change with “mechanical strength [6,7]”
“and puberty (Figure 1)”, change with “and restoration (Figure 1)”, which is consistent with Figure 1, or “and regeneration (Figure 1)”.
The text between Figure 1 and 2 is not justified at the margins.
Section 2.
“ (300 and 58%, respectively)” change with “(300% and 58%, respectively)”
PCL acronym is not given in extenso.
Section 3.1
“time. , The” correct the punctuation typo.
“[38].The” a blank space is needed after punctuation.
Figure 3. Legend: if this is taken from a paper, check copyright permission, cite them and the reference in the legend.
Table 1: Full names of solvents and polymers must be given in the table legend or table note.
Section 3.2
“CaCO3” (also in the table), the number 3 is under script.
“poly (lactic acid)” add acronym here “poly (lactic acid) (PLA)”.
“(Kim, Lee, et al. 2006)[48]” this reference must be corrected.
“natural polymers (e.g., gelatin) And Silk”, change with “natural polymers (e.g., Gt) and silk”
“PCB / nHA /” never given in extenso the first time mentioned
Table 2: Full names of solvents and polymers must be given in the table legend or table note.
“CNT” never given in extenso the first time mentioned.
Please check all acronyms.
Author Response
Thank you very much for your constructive suggestions in “Manuscript ID: membranes-1349131” on 31 August 2021. Based on editor and reviewer’s comments and suggestions, the changes within the revised manuscript have been highlighted and our responses are responded in the attachment file.
